# Patients Diagnosed with Granulomatosis with Polyangiitis: The Journey to Receive Rheumatologist Care

**DOI:** 10.3390/jcm14103523

**Published:** 2025-05-18

**Authors:** Nihal Lermi, Burcu Yağız, Ali Ekin, Belkıs Nihan Coşkun, Ediz Dalkılıç, Yavuz Pehlivan

**Affiliations:** 1Department of Rheumatology, Harakani Public Hospital, Kars 36200, Turkey; 2Division of Rheumatology, Faculty of Medicine, Uludag University, Bursa 16059, Turkey; burcuyilmaz_84@hotmail.com (B.Y.); belkisnihanseniz@hotmail.com (B.N.C.); edizinci@hotmail.com (E.D.); drypehlivan@gmail.com (Y.P.); 3Division of Rheumatology, Van Yüzüncü Yıl University Faculty of Medicine, Van 65090, Turkey; aliekin49@hotmail.com

**Keywords:** granulomatosis with polyangiitis, clinical presentation, diagnostic delay, early diagnosis

## Abstract

**Background/Objectives**: Granulomatosis with polyangiitis (GPA) is a necrotising vasculitis characterised by granulomatous inflammation involving small vessels. In addition to specific findings for the affected organ, constitutional symptoms and joint and muscle pain can be observed. The prodromal phase, where symptoms last for months before clinical diagnosis, may suggest infection or malignancy. This may cause a delay in GPA diagnosis. The period from the first symptoms to diagnosis may last from one month to three years. In this study, we aimed to demonstrate that, as the time between the onset of symptoms and diagnosis increases, the disease involvement may become more severe, and the possibility of recurrence may increase, indicating the importance of early diagnosis. **Methods**: For this cross-sectional retrospective study, data from 40 patients with GPA were evaluated. Demographic, clinical, treatment, and follow-up characteristics of the patients were obtained from their medical records. **Results**: The mean time between the presentation of the first complaints and GPA diagnosis was 7.40 ± 11.84 (0, 60, 3; mean ± standard deviation [minimum, maximum, and median]) months. The time between the first complaints and diagnosis was longer for patients with a history of recurrence (11.44 ± 16.73 [0, 60, 4.5] vs. 4.71 ± 6.04 [0, 24, 2.5], *p* value = 0.260). **Conclusions**: GPA is an inflammatory disease with various clinical presentations. In the management of patients with GPA, awareness of its presentation is important for rheumatologists as well as other clinicians during the initial evaluation, demonstrating the importance of interclinical collaboration.

## 1. Introduction

Granulomatosis with polyangiitis (GPA) is a necrotising autoimmune condition that is categorised as antineutrophil cytoplasmic antibody (ANCA)-associated vasculitis and involves small vessels (capillaries, venules, arterioles and small arteries, and veins) [1,2,3]. GPA, a rare disease, is observed equally in both sexes, and the average age of onset is 45–65 years [2].

Diagnosis of GPA is made with ANCA positivity in the presence of clinical findings, histological demonstration of necrotising vasculitis, necrotising glomerulonephritis, and granulomatous inflammation in biopsy from related tissue, such as skin, lung, or kidney [4]. In GPA, there is an association of vascular wall inflammation and peri- and extravascular granulomatosis. The predominant ANCA pattern is cytoplasmic, while the predominant autoantigen is leukocyte proteinase 3 (PR3), which is produced by neutrophils and plays a key role in pathogenesis [3,5,6].

GPA can be systemically or locally involved. In addition to the specific findings for the affected organ, constitutional symptoms, such as fatigue, weight loss, fever, migratory joint pain, and muscle pain, can be observed [3]. The disease is also classified as severe or nonsevere. The type of involvement and the presence of severe disease determine the treatment method.

The prodromal phase, where symptoms persist for months before clinical diagnosis, may suggest infection or malignancy, which are more frequently considered and more common. This may cause a delay in GPA diagnosis [6]. It has been reported in the literature that the period from the first symptoms to diagnosis can range from one month to three years [7,8,9]. Hence, the cooperation of rheumatologists and other physicians that the patient will consult before reaching a rheumatologist and the awareness of these physicians regarding GPA are important.

In this study, we aimed to demonstrate that as the time between symptom onset and diagnosis increases, disease involvement may become more severe, and the likelihood of relapse may increase. Hence, early diagnosis is important.

## 2. Materials and Methods

This retrospective single-centre study was conducted with the approval of the ethics committee of Bursa Uludağ University Faculty of Medicine (approval no. 2024/9-10, dated 5 June 2024) and in accordance with the principles of the Declaration of Helsinki. The data of 49 patients who received follow-up care with GPA diagnosis in the Rheumatology Department Outpatient Clinic of Bursa Uludağ University Faculty of Medicine between January 2010 and March 2024 were examined by scanning the hospital electronic file system. Nine patients whose data could not be accessed were excluded from this study; therefore, 40 patients were included in this study. The inclusion criteria for this study were having a GPA diagnosis, receiving follow-up care in the Rheumatology outpatient clinic, and being over 18 years of age. Patients were referred to a rheumatologist by the physicians who initially evaluated them based on the following criteria: persistent constitutional symptoms and rash unresponsive to treatment; upper respiratory tract involvement or otitis media not responding to therapy; concurrent kidney and lung involvement; cavitary lung lesions identified through imaging; biopsy findings from any affected organ consistent with ANCA-associated vasculitis; laboratory evidence of ANCA positivity; and rheumatologic complaints such as arthritis. When patients presented to us, GPA classification was based on the 1990 ACR criteria and/or the 2022 ACR/EULAR criteria. The disease activity was assessed at the time of diagnosis or during follow-up visits by using the Birmingham Vasculitis Activity Score (BVAS). The lungs of patients with respiratory symptoms were auscultated by their physicians at the first visit, last visit, and during follow-up visits. Chest X-ray and, if necessary, computed tomography were used as imaging modalities. The ANCA tests of our patients were performed using indirect immunofluorescence (IIF) and enzyme-linked immunosorbent assay (ELISA) methods.

The following parameters were collected from the records: (1) demographic findings (sex, age, age at diagnosis, and smoking at the time of diagnosis); (2) questioning about the symptoms that caused the patient to visit a doctor for the first time before diagnosis (hearing loss, another symptom related to the ear, sinusitis, respiratory or constitutional symptoms, eye complaints, joint pain, skin rash, facial paralysis, foot drop, stenosis symptoms related to the glottis or trachea [stridor], and/or symptoms related to the pituitary gland [frequent urination, narrowing of the visual field]); (3) first outpatient clinic visited (Ear, Nose, and Throat [ENT], Chest Diseases, Ophthalmology, Internal Medicine, Infectious Diseases and Microbiology, Chest Surgery, Nephrology, Dermatology, Rheumatology, or Neurosurgery); (4) time between first outpatient clinic visit and GPA diagnosis (months); (5) involvement at the time of diagnosis (lungs, auditory system, ear, nasal, sinus, glottis, tracheal, eye, kidney, joint, skin, gastrointestinal, and/or neurological involvement) and BVAS results of patients at the time of diagnosis; (6) serological anti-PR3 (cytoplasmic ANCA) and myeloperoxidase (MPO [perinuclear-ANCA] positivity); (7) organ or region of biopsy when diagnosing GPA (nose, lungs, bronchoalveolar lavage, kidneys, eyes, pituitary adenoma extraction, etc.); (8) induction therapy provided at the time of diagnosis (corticosteroid [CS], cyclophosphamide [Cyc], rituximab [Rtx], azathioprine [AZA], methotrexate [Mtx], or mycophenolate mofetil [MMF]); (9) treatment agent used for maintenance (CS, Cyc, Rtx, AZA, Mtx, MMF, plasmapheresis, or intravenous immunoglobulin [IVIG]); (10) disease duration at the last visit (months); (11) presence of disease recurrence at the last visit; (12) presence of complaints at the last visit (hearing problems, ear symptoms, nasal symptoms, sinusitis, respiratory symptoms, visual or joint-related symptoms, and/or skin rash); (13) laboratory values checked during the examination (including protein in the urine and pathology in the postero-anterior radiograph); and (14) treatment administered during the examination (CS, Cyc, Rtx, AZA, Mtx, MMF, or plasmapheresis).

IBM SPSS 24 software for Windows (SPSS Inc., Chicago IL, USA) was used for statistical analyses. Descriptive statistics are presented as mean ± standard deviation and median (minimum–maximum) values for measured variables and as frequencies and percentages (%). Categorical variables of patients with and without a history of recurrence were compared using Pearson’s chi-square and Fisher’s exact tests, as appropriate. A *p* value < 0.05 was considered statistically significant.

## 3. Results

This study included 40 patients diagnosed with GPA, of whom 21 (52.5%) were female. The mean age of the patients was 49.05 ± 11.53 (25, 73, 47.5; mean ± standard deviation [minimum, maximum, and median]) years, and the mean age at diagnosis was 43.2 ± 11.66 (21, 68, 43.5) years. Of the 17 patients whose smoking history data were available, 10 (58%) were smokers at the time of diagnosis.

When the first symptoms of the patients were examined, joint pain was present in 27 (67.5%), respiratory symptoms in 20 (50%), ear symptoms in 19 (47.5%), nasal symptoms in 18 (45%), constitutional symptoms in 18 (45%), and hearing loss in 14 (35.5%) patients (Table 1). The first outpatient clinics that the patients visited (Table 2) most frequently were ENT (14 patients, 35%) and Chest Diseases (10 patients, 25%). The mean time between the presentation of the first complaints and the determination of a GPA diagnosis was 7.40 ± 11.84 (0, 60, and 3) months.

At the time of diagnosis, lung involvement was detected in 27 (67.5%), joint involvement in 27 (67.5%), auditory system involvement in 14 (35%), nasal involvement in 19 (47.5%), and kidney involvement in 17 (42.5%) patients (Table 1). The mean BVAS result of patients at the time of diagnosis was 12.97 ± 5.98 (3, 13, and 32). There were 32 (80%) patients with PR3 positivity and 1 (2.5%) patient with MPO positivity. GPA diagnosis was supported by biopsy in 23 (57.5%) patients. Findings consistent with GPA were detected in kidney samples of eight (20%) patients, lung samples of seven (17.5%) patients, nasal cavity samples of four (10%) patients, bronchoalveolar lavage samples of two (5%) patients, eye sample of one (2.5%) patient, and pituitary biopsy sample of one (2.5%) patient.

The induction treatment that was administered to patients at the time of diagnosis, maintenance treatment provided during follow-up care, and the treatment options of patients at the last examination are shown in Table 3. The treatments provided at the time of diagnosis were most frequently CS for 40 (100%) and Cyc for 32 (80%) patients. The maintenance treatment options provided were most frequently CS for 37 (92.5%) and Rtx for 30 (75%) patients. The treatments provided at the last visit were most frequently CS for 35 (87.5%) and Rtx for 23 (57.5%) patients.

The mean disease duration of the patients recorded at the last visit was 71.3 ± 48.41 (1, 166, 67.5) months. There were 16 (40%) patients who developed disease recurrence during the follow-up period. At the last visit, 12 (30%) patients had hearing loss, 7 (17.5%) had ear symptoms, 8 (20%) had nasal symptoms, 1 (2.5%) had sinusitis symptoms, 2 (5%) had respiratory symptoms, 1 (2.5%) had ocular symptoms, and 4 (10%) had joint pain complaints. No patient had a skin rash. At the last visit, 11 (27.5%) patients had proteinuria, as ascertained by a complete urine test, and 7 (17.5%) patients had pathological findings in their chest X-ray.

The characteristics of patients who exhibited and did not exhibit recurrence during follow-up visits are shown in Table 4. There was no statistically significant difference between age and the age at diagnosis of patients who developed and did not develop recurrence (*p* values = 0.523 and 0.803). Although it did not reach statistical significance, the duration between the first visit to the doctor and GPA diagnosis was longer in patients with a history of recurrence (11.44 ± 16.73 [0, 60, 4.5] vs. 4.71 ± 6.04 [0, 24, 2.5], *p* value = 0.260). The disease duration recorded at the last visit was statistically significantly higher for patients who developed recurrence (99.69 ± 48.59 [9, 166, 103.5] vs. 52.38 ± 38.73 [1, 156, 50], *p* value = 0.002). There was no statistically significant difference between sex, PR3 and MPO positivity, BVAS results of patients at the time of diagnosis, biopsy at diagnosis, symptoms at diagnosis, and involvement at diagnosis for patients who developed and did not develop recurrence (*p* > 0.005). While there was no statistically significant difference between the use of Mtx, AZA, and Cyc at diagnosis of patients who developed and did not develop recurrence (*p* values = 0.588, 0.408, and 0.872), there was a statistically significant difference between the use of Rtx and plasmapheresis (*p* values = 0.027 and 0.030).

## 4. Discussion

In this study, we examined the diagnosis, access to a rheumatologist, and treatment processes of patients with GPA after their first complaints. In patients with disease relapse during their follow-up visits, the time between the onset of complaints and GPA diagnosis was longer, and they had a longer disease duration. We emphasise the importance of collaboration with other clinics and other physicians’ (for example, ENT and Chest Diseases) awareness of GPA symptoms. Rtx was used less as an induction treatment for the group of patients with relapse.

GPA may present with various symptoms, with systemic (74.1%), ENT (71.9%), and pulmonary symptoms (67.4%) being the most common [10]. Muscle and bone symptoms are observed in approximately 50% of patients. Polyarthritis is usually detected. Common ENT symptoms are chronic sinusitis, rhinitis, epistaxis, and deafness. Antibiotic-resistant otitis media, nasal septum perforation, nasal bone deformation, and, more rarely, subglottal stenosis and unilateral exophthalmos can be observed. Lower respiratory tract symptoms are nonspecific. Cough, dyspnoea, chest pain, and haemoptysis can be observed [3,6,11]. Webb et al. noted that ENT specialists play an important role in the evaluation of patients with GPA and that ENT physicians’ awareness of other presentations of GPA can contribute significantly to rapid diagnosis and treatment [10]. Greco et al. emphasised that otologic symptoms may be the first symptoms in patients with GPA and that ENT specialists should suspect GPA in cases of otitis media that do not improve with long-term antibiotic treatment [4]. The first symptoms of our patients were joint pain (67.5%), respiratory symptoms (50%), ear symptoms (47.5%), nasal symptoms (45%), constitutional symptoms (45%), and hearing loss (35.5%).

The awareness of physicians in other clinics of GPA is important for diagnosis. In Pearce et al.’s study, 9.2% of patients with GPA were evaluated in an ENT clinic in the one year before diagnosis, compared with 1.4% in the general population. This study emphasised the importance of the ENT clinic as an opportunity to identify these patients [12]. The first outpatient clinics that our patients visited were ENT (35%) and Chest Diseases (25%). We believe our patients visited a physician less frequently due to less specific joint pain and constitutional symptoms and more frequently for more specific ENT and respiratory symptoms.

Several factors may cause a delay in GPA diagnosis. In addition to specific findings for the affected organ in GPA, constitutional symptoms and migratory joint complaints are frequently observed. These findings may occur weeks or months before organ involvement. During this period, due to similar symptoms, patients and physicians may be concerned about infection or malignancy, and patients may be examined in this direction [5,6]. The variety and nonspecificity of symptoms can often cause delays in diagnosis, which can result in increased mortality and morbidity [13,14,15]. In addition, GPA may begin with the involvement of only the upper respiratory tract without general symptoms, and ANCA is detected in only some of these patients. Biopsy yield is also lower in biopsies taken from the upper respiratory tract. Recognition of the disease may become difficult at this stage and may cause delays in diagnosis [5,6,11,16]. Kronbichler et al. emphasised that symptoms related to the organ involved in presentation are effective for addressing diagnostic difficulty and delays in diagnosis. The authors mentioned that a patient presenting with slowly progressing symptoms concerning the upper respiratory tract is more likely to present diagnostic difficulties than a patient admitted with rapidly progressing pulmonary renal syndrome [16]. In Jacyshyn et al.’s study, the time from the first symptoms to diagnosis was reported as one month to three years, and 36% of the symptoms started more than one year before diagnosis [8]. Labrador et al. found the symptom onset before diagnosis to be 6.5 months [9]. Abdou et al. reported that only 22% of patients were diagnosed within the first month of symptom onset, while 46% were diagnosed one to six months later. They showed that 15% of patients had a diagnosis delay of 6–12 months, and 18% had a delay of more than 12 months [7]. The mean time between our patients’ first complaints and their GPA diagnosis was 7.40 ± 11.84 (0, 60, 3) months.

In GPA, upper respiratory tract involvement is observed in 70–100% of patients and in the early stages of the disease. The nasal cavity and paranasal sinuses are more frequently affected. Sinonasal, otological, and tracheal symptoms can be observed in 87–93% of patients with GPA throughout the course of the disease [10]. Respiratory system involvement is common in patients with GPA and is generally in the form of parenchymal involvement (85%). Solitary or multiple nodules of different sizes, with accompanying cavities, can be observed. Asymptomatic lung involvement can be found in 30% of patients. In these cases, alveolar haemorrhage and tracheobronchial involvement may be detected later. Renal involvement is associated with high mortality and morbidity and can be observed in 70–85% of patients with GPA throughout the course of the disease. Therefore, it is important to control haematuria and proteinuria at each visit [3,5,6]. Skin, eye, orbit, neurological, and, less frequently, gastrointestinal and breast involvement may be observed [2,4,17]. In our patients, lung involvement was detected in 67.5%, joint involvement in 67.5%, auditory system involvement in 35%, nasal involvement in 47.5%, and renal involvement in 42.5% at the time of diagnosis.

GPA diagnosis is based on a combination of clinical findings, imaging, laboratory (erythrocyte sedimentation rate and c-reactive protein, complete blood count, and renal and urine parameters) findings, serological markers (ANCA testing), and histopathological findings when biopsy is possible [5,18]. ANCA is an important definitive marker in patients with ANCA-associated vasculitis (AAV). This antibody test should be ordered for all patients with clinical suspicion of vasculitis. ANCA is not specific for all AAV, and a negative ANCA test does not exclude AAV. PR3 ANCA positivity in GPA is between 65% and 75% [2]. Thirty-two (80%) of our patients had a positive PR3 result. Radiology plays an important role in the diagnosis and follow-up care of patients with GPA and helps to differentiate GPA from other diseases that may mimic it. Soto et al. emphasised that radiologists can guide GPA diagnosis with imaging, as radiology can help distinguish between disease activity and chronic damage, and this guidance can affect treatment decisions [11].

Biopsy is the gold standard for GPA diagnosis [19]. In GPA, there is a combination of vascular wall inflammation and peri- and extravascular granulomatosis. However, biopsy is not applicable to every patient, and treatment can start before obtaining the biopsy result. Biopsy is more important in ANCA-negative patients and limited forms of GPA. The efficiency of biopsy in GPA diagnosis may vary depending on the region where it is taken and the amount of tissue sample taken, and failure to demonstrate histological features does not exclude GPA diagnosis. Although the upper respiratory tract is easily accessible, 24% of biopsies from this region support GPA diagnosis [3,5,6]. Moreover, although head and neck biopsy has low specificity for GPA, it is useful in excluding malignancies such as squamous cell carcinoma and T-cell lymphoma and infections from, for example, fungal agents [10]. Renal biopsy should be performed promptly in cases of abnormal urine sediment, proteinuria, and elevated serum creatinine. Direct findings for crescentic glomerulonephritis (pauci-immune necrotising glomerulonephritis) and vasculitis in the vasa recta may be detected. In addition, kidney biopsy can provide information about parenchymal involvement, fibrotic lesions, and tubular necrosis, thus providing insight into prognosis and the possibility of disease reversal [3,20]. In our study, GPA diagnosis was supported via biopsy in 23 (57.5%) patients. Findings consistent with GPA were detected in eight (20%) patients’ kidneys, seven (17.5%) patients’ lungs, four (10%) patients’ nasal cavities, two (5%) patients’ bronchoalveolar lavage samples, one (2.5%) patient’s eye, and one (2.5%) patient’s pituitary biopsy.

GPA treatment is based on the combination of CS and immunosuppressive agents. In systemic forms, CSs alone are insufficient to induce and maintain remission. In severe forms, the combination of CS and Cyc or Rtx is the standard treatment for induction of remission. However, there is increasing evidence that Rtx is a preferred remission induction regimen in certain patient subgroups, such as PR3-ANCA-positive patients, those with GPA relapse, and children and adults in whom fertility preservation is important. In nonsevere forms, Rtx for remission induction is increasingly gaining ground. Mtx and MMF are alternative treatments in cases where Rtx or Cyc cannot be used. Rtx and low-dose CSs are at the forefront in maintenance treatment. In cases where Rtx is contraindicated, AZA, Mtx, and to a lesser extent, MMF can be preferred. Plasma exchange was previously a preferred treatment method in cases of rapidly progressive severe glomerulonephritis and alveolar haemorrhage, but its role in treatment is controversial today. IVIG may be a choice in recurrent disease [2,3,6,21,22,23,24]. In our study, the treatments administered at the time of diagnosis were CS for 40 (100%), Cyc for 32 (80%), and Rtx for three (7.5%) patients. The treatments administered as maintenance were CS for 37 (92.5%), Rtx for 30 (75%), AZA for 13 (32.5%), and Mtx for six (15%) patients, consistent with the literature. IVIG maintenance treatment was administered to one (2.5%) patient due to refractory ocular disease. The most common treatments provided to the patients at the last visit were CS for 35 (87.5%) and Rtx for 23 (57.5%) patients.

In untreated GPA cases, the one-year mortality rate is approximately 70%. With treatment, remission rates exceed 80% of cases. Relapses are common and occur in more than 50% of cases. However, the 10-year survival rate is 75%. Anti-PR3 ANCA positivity at diagnosis, cardiac involvement, and creatinine clearance < 60 mL/min are factors associated with relapse [3,25]. ENT involvement has a better prognosis but is associated with more frequent relapses. The probability of recurrence is higher in patients with lung involvement, nasal Staphylococcus aureus colonisation, and a history of previous recurrence [3,6]. Patients should be closely monitored for recurrence after induction therapy. Recurrence symptoms may not be as obvious as the initial findings for the disease. When recurrence occurs, mimics of the primary diagnosis (vasculitis), such as infection, malignancy, and intravenous drug use, should be excluded. Recurrence treatment is similar to primary treatments, and there are studies indicating that Rtx is superior [2,26]. The mean disease duration recorded at the last visit was 71.3 ± 48.41 (1, 166, 67.5) months. There were 16 patients (40%) who developed disease recurrence during their follow-up visits. Although it did not reach statistical significance in patients with a history of recurrence, the time between the patients’ first visit to the doctor and GPA diagnosis was longer. The disease duration at the last visit was statistically significantly longer in patients with recurrence. While there was no statistically significant difference between the use of Mtx, AZA, and Cyc at the time of diagnosis in patients with and without recurrence, there was a statistically significant difference between the use of Rtx and plasmapheresis. All 24 patients who did not relapse received Rtx treatment at the time of diagnosis. Thirteen of the sixteen patients who relapsed (81.3%) had no history of Rtx use at the time of diagnosis. Although there was a statistically significant difference, the number of patients is insufficient to conclude that the use of Rtx in induction therapy can prevent relapse. However, this result supports studies demonstrating that Rtx is superior for induction treatment [24].

One of the limitations of our study is the relatively small sample size, which may limit the generalisability of the findings. We acknowledge that a larger sample size could provide more robust results and offer a better understanding of GPA’s characteristics and progression.

## 5. Conclusions

GPA is a multisystem inflammatory disease with various clinical presentations. Early diagnosis and other physicians’ awareness of GPA presentation are necessary for successful treatment. Initial evaluation by rheumatologists as well as other clinicians, such as ENT and Chest Disease specialists, plays an important role in the management of patients with GPA. Therefore, interclinical collaboration is crucial in the treatment of GPA.

## Figures and Tables

**Table 1 jcm-14-03523-t001:** The symptoms that caused the patient to visit a doctor for the first time before diagnosis and involvement at the time of diagnosis.

The Symptoms		No *n* (%)	Yes *n* (%)
	Hearing loss	26 (65%)	14 (35%)
	Ear	21 (52.5%)	19 (47.5%)
	Nasal	22 (55%)	18 (45%)
	Sinusitis	29 (72.5%)	11 (27.5%)
	Related to the glottis or trachea	38 (95%)	2 (5%)
	Respiratory symptoms	20 (50%)	20 (50%)
	Constitutional symptoms	22 (55%)	18 (45%)
	Eye complaints	29 (72.5%)	11 (27.5%)
	Joint pain	13 (32.5%)	27 (67.5%)
	Skin rash	30 (75%)	10 (25%)
	Facial paralysis	37 (92.5%)	3 (7.5%)
	Foot drop	39 (97.5%)	1 (2.5%)
	Vasculitic findings	35 (87.5%)	5 (12.5%)
	Related to the pituitary gland	39 (97.5%)	1 (2.5%)
Involvement			
	Lungs	13 (32.5%)	27 (67.5%)
	Auditory system	26 (65%)	14 (35%)
	Ear	22 (55%)	18 (45%)
	Nasal	21 (52.5%)	19 (47.5%)
	Sinus	29 (72.5%)	11 (27.5%)
	Glottis tracheal	38 (95%)	2 (5%)
	Eye	31 (77.5%)	9 (22.5%)
	Kidney	23 (57.5%)	17 (42.5%)
	Joint	13 (32.5%)	27 (67.5%)
	Skin	30 (75%)	10 (25%)
	Gastrointestinal	37 (92.5%)	3 (7.5%)
	Neurological	37 (92.5%)	3 (7.5%)

**Table 2 jcm-14-03523-t002:** First outpatient clinic visited.

First Outpatient Clinic Visited	*n* (%)
Ear, Nose, and Throat	14 (35%)
Chest Diseases	10 (25%)
Rheumatology	5 (12.5%)
Ophthalmology	2 (5%)
Internal Medicine	2 (5%)
Infectious Diseases and Microbiology	2 (5%)
Nephrology	2 (5%)
Chest Surgery	1 (2.5%)
Dermatology	1 (2.5%)
Neurosurgery	1 (2.5%)

**Table 3 jcm-14-03523-t003:** Treatments administered during the induction, maintenance, and last examination.

Treatment Agent	Induction (n: 40)	Maintenance (n: 40)	Examination (n: 39)
CS	40 (100%)	37 (92.5%)	35 (87.5%)
Mtx	6 (15%)	10 (25%)	5 (12.57%)
AZA	1 (2.5%)	13 (32.5%)	7 (17.5%)
Cyc	32 (80%)	6 (15%)	3 (7.5%)
Rtx	3 (7.5%)	30 (75%)	23 (57.5%)
Plasmapheresis	6 (15%)	1 (2.5%)	
MMF		2 (5%)	1 (2.5%)
IVIG		1 (2.5%)	
Only CS	2 (5%)		2 (5.1%)
Only Mtx			1 (2.6%)
Only Rtx			2 (5.1%)
Cyc + CS	22 (55%)		3 (7.6%)
Cyc + CS + AZA	1 (2.5%)		
Cyc + CS + plasmapheresis	6 (15%)		
Cyc + Cs + Rtx	3 (7.5%)		
CS + Mtx	6 (15%)		3 (7.6%)
CS + AZA			6 (15.3%)
CS + Rtx			19 (48.7%)
CS + MMF			1 (2.6%)
CS + AZA + Rtx			1 (2.6%)
Rtx + Mtx			1 (2.6%)

Azathioprine (AZA), corticosteroid (CS), cyclophosphamide (Cyc), intravenous immunoglobulin (IVIG), methotrexate (Mtx), mycophenolate mofetil (MMF), or rituximab (Rtx).

**Table 4 jcm-14-03523-t004:** The characteristics of patients who exhibited and did not exhibit recurrence during follow-up visits and the whole group.

		Whole Group (n = 40)			Recurrence No (n = 24)			Recurrence Yes (n = 16)			*p* Value *
		Mean ± SD/ n %		Median	Mean ± SD/ n %		Median	Mean ± SD/ n %		Median	
CHARACTERISTICS OF PATİENTS											
Age		49.05 ± 11.53		47.5	48.08 ± 10.68		47.5	50.5 ± 12.91		48.5	0.523 ^t^
Age at diagnosis		43.2 ± 11.66		43.5	43.58 ± 11.75		43.5	42.63 ± 11.89		43.5	0.803 ^t^
The duration between the first visit and GPA diagnosis (months)		7.40 ± 11.84		3	4.71 ± 6.04		2.5	11.44 ± 16.73		4.5	0.260 ^t^
Disease duration as of the last visit (months)		71.3 ± 48.41		67.5	52.38 ± 38.73		50	99.69 ± 48.59		103.5	0.002 ^t^
BVAS results of patients at the time of diagnosis		12.97 ± 5.98		12	11.83 ± 5.01		10	14.69 ± 7.02		15	0.179 ^m^
Sex	Female	21	52.5%		12	50%		9	56.3%		0.698 ^x2^
	Male	19	47.5%		12	50%		7	43.8%		
PR3 positivity	No	8	20%		4	16.7%		4	25%		0.519 ^x2^
	Yes	32	80%		20	83.3%		12	75%		
MPO positivity	No	39	97.5%		23	95.8%		16	100%		0.408 ^x2^
	Yes	1	2.5%		1	4.2%					
Biopsy at diagnosis	No	17	42.5%		10	41.7%		7	43.8%		0.968 ^x2^
	Yes	23	57.5%		14	58.3%		9	56.3%		
THE SYMPTOMS THAT CAUSED THE PATIENT TO VISIT A DOCTOR											
Hearing loss	No	26	65%		18	75%		8	50%		0.104 ^x2^
	Yes	14	35%		6	25%		8	50%		
Ear	No	21	52.5%		15	62.5%		6	37.5%		0.121 ^x2^
	Yes	19	47.5%		9	37.5%		10	62.5%		
Nasal	No	22	55%		14	58.3%		8	50%		0.604 ^x2^
	Yes	18	45%		10	41.7%		8	50%		
Sinusitis	No	29	72.5%		18	75%		11	68.8%		0.665 ^x2^
	Yes	11	27.5%		6	25%		5	31.3%		
Related to the glottis or trachea	No	38	95%		23	95.8%		15	93.8%		0.767 ^x2^
	Yes	2	5%		1	4.2%		1	6.3%		
Respiratory symptoms	No	20	50%		13	54.2%		7	43.8%		0.519 ^x2^
	Yes	20	50%		11	45.8%		9	56.3%		
Constitutional symptoms	No	22	55%		14	58.3%		8	50%		0.604 ^x2^
	Yes	18	45%		10	41.7%		8	50%		
Eye complaints	No	29	72.5%		20	83.3%		9	56.3%		1.060 ^x2^
	Yes	11	27.5%		4	16.7%		7	43.8%		
Joint pain	No	13	32.5%		5	20.8%		8	50%		0.054 ^x2^
	Yes	27	67.5%		19	79.2%		8	50%		
Skin rash	No	30	75%		19	79.2%		11	68.8%		0.456 ^x2^
	Yes	10	25%		5	20.8%		5	31.3%		
Facial paralysis	No	37	92.5%		22	91.7%		15	93.8%		0.806 ^x2^
	Yes	3	7.5%		2	8.3%		1	6.3%		
Foot drop	No	39	97.5%		23	95.8%		16	100%		1.408 ^x2^
	Yes	1	2.5%		1	4.2%					
Vasculitic findings	No	35	87.5%		21	87.5%		14	87.5%		1.000 ^x2^
	Yes	5	12.5%		3	12.5%		2	12.5%		
Related to the pituitary gland	No	39	97.5%		24	100%		15	93.8%		0.215 ^x2^
	Yes	1	2.5%					1	6.3%		
INVOLVEMENT AT THE DIAGNOSIS											
Lungs	No	13	32.5%		9	37.5%		4	25%		0.408 ^x2^
	Yes	27	67.5%		15	62.5%		12	75%		
Auditory system	No	26	65%		18	75%		8	50%		0.104 ^x2^
	Yes	14	35%		6	25%		8	50%		
Ear	No	22	55%		16	66.7%		6	37.5%		0.069 ^x2^
	Yes	18	45%		8	33.3%		10	62.5%		
Nasal	No	21	52.5%		14	58.3%		7	43.8%		0.366 ^x2^
	Yes	19	47.5%		10	41.7%		9	56.3%		
Sinus	No	29	72.5%		18	75%		11	68.8%		0.665 ^x2^
	Yes	11	27.5%		6	25%		5	31.3%		
Glottis tracheal	No	38	95%		23	95.8%		15	93.8%		0.767 ^x2^
	Yes	2	5%		1	4.2%		1	6.3%		
Eye	No	31	77.5%		20	83.3%		11	68.8%		0.600 ^x2^
	Yes	9	22.5%		4	16.7%		5	31.3%		
Kidney	No	23	57.5%		13	54.2%		10	62.5%		0.601 ^x2^
	Yes	17	42.5%		11	45.8%		6	37.5%		
Joint	No	13	32.5%		5	20.8%		8	50%		0.054 ^x2^
	Yes	27	67.5%		19	79.2%		8	50%		
Skin	No	30	75%		19	19%		11	68.8%		0.456 ^x2^
	Yes	10	25%		5	5%		5	31.3%		
Gastrointestinal	No	37	92.5%		22	91.7%		15	93.8%		0.806 ^x2^
	Yes	3	7.5%		2	8.3%		1	6.3%		
Neurological	No	37	92.5%		22	91.7%		15	93.8%		0.806 ^x2^
	Yes	3	7.5%		2	8.3%		1	6.3%		
INDUCTION THERAPY											
CS	No	0	0		0			0			
	Yes	40	100%		24	100%		16	100%		
Mtx	No	34	85%		21	87.5%		13	81.3%		0.588 ^x2^
	Yes	6	15%		3	12.5%		3	18.8%		
AZA	No	39	97.5%		23	95.8%		16	100%		0.408 ^x2^
	Yes	1	2.5%		1	4.2%		0			
Cyc	No	8	20%		5	20.8%		3	18.8%		0.872 ^x2^
	Yes	32	80%		19	79.2%		13	81.2%		
Rtx	No	13	32.5%		0			13	81.3%		0.027 ^x2^
	Yes	27	67.5%		24	100%		3	18.8%		
Plasmapheresis	Yes	34	85%		18	75%		16	100%		0.030 ^x2^
	No	6	15%		6	25%		0			

Azathioprine (AZA), Birmingham Vasculitis Activity Score (BVAS), corticosteroid (CS), cyclophosphamide (Cyc), leukocyte proteinase 3 (PR3), methotrexate (Mtx), myeloperoxidase (MPO), or rituximab (Rtx). * The *p* value indicates the statistical significance between the relapsing and non-relapsing groups. ^x2^ Chi-square test/^m^; Mann–Whitney U test/^t^ t-test.

## Data Availability

The data of the patients’ medical history will be provided by the authors without hesitation if requested.

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
