# Peer review of "Patients Diagnosed with Granulomatosis with Polyangiitis: The Journey to Receive Rheumatologist Care"

_jcm, 2025, doi:10.3390/jcm14103523_

Round 1
Reviewer 1 Report
Comments and Suggestions for Authors
- I don’t think the manuscript title does not reflect the entire content. The clinical courses how the patients were referred to rheumatology department are not introduced in detail. The authors discussed reasons of relapses later.
- While the study includes MPO-ANCA positive GPA, MPO-ANCA positivity is a negating factor for GPA in the 2022 ACR/EULAR criteria. What criteria were applied for the classification of GPA in this study?
- It is desirable to show entire patient demographics before proceeding into comparison. The authors stated they assessed BVAS in materials and methods section. Where can I find the results?
- Table 1: the percentage of ‘Hearing loss Yes’ would be 35.0 %. Methods to detect respiratory symptoms in this study such as auscultation, chest X-ray and computed tomography, should be described. And methods to detect ANCA in this study such as ELISA, and Ouchterlony, should be described.
- Table 3: it is desirable to also show information on multi-targeted therapy.
- Table 4: ‘Recurrens’ would be spelling errors of ‘recurrence’. When seeing ‘INDUCTION THERAPY’ ‘CS’, the sum of patients numbers is not 40. No patients received RTX in the group WITHOUT relapses, whereas a couple of patients received RTX in the group with relapses?
Please refer to Comments for Authors.
Author Response
"Please see the attachment."

Reviewer 2 Report
Comments and Suggestions for Authors
In this article, the authors aimed to offer a detailed retrospective analysis of diagnostic delays in Granulomatosis with Polyangiitis and their correlation with disease severity and recurrence.
Objectives and Design are well defined. The data collection is comprehensive, including symptoms, initial clinic visits, serology, biopsy findings, and treatment details.
Any study requires the informed consent of the individuals. You have not mentioned anything about this information.
Tables are used effectively to organize data.
I recommend that you ask for the help of a native speaker or an authorized translator.
to revise the grammar and clarity (e.g. line 8 - "Correspondence: Correspondence: ", typos).
I recommend to add discussion on limitations, especially related to sample size.
The article presents 26 references being up to date.
Author Response
"Please see the attachment."

Round 2
Reviewer 1 Report
Comments and Suggestions for Authors
(No further comments)